# Patient-Controlled Intravenous Morphine Analgesia Combined with Transcranial Direct Current Stimulation for Post-Thoracotomy Pain: A Cost-Effectiveness Study and A Feasibility for Its Future Implementation

**DOI:** 10.3390/ijerph17030816

**Published:** 2020-01-28

**Authors:** Nemanja Rancic, Katarina Mladenovic, Nela V. Ilic, Viktorija Dragojevic-Simic, Menelaos Karanikolas, Tihomir V. Ilic, Dusica M. Stamenkovic

**Affiliations:** 1Medical Faculty Military Medical Academy, University of Defense, 11 000 Belgrade, Serbia; nece84@hotmail.com (N.R.); katarina.mladenovic72@gmail.com (K.M.); vdragsim@gmail.com (V.D.-S.); tihoilic@gmail.com (T.V.I.); 2Center for Clinical Pharmacology, Military Medical Academy, 11 000 Belgrade, Serbia; 3Institute of Radiology, Military Medical Academy, 11 000 Belgrade, Serbia; 4Department of Anesthesiology and Intensive Care, Military Medical Academy, 11 000 Belgrade, Serbia; 5Medical Faculty, University of Belgrade, 11 000 Belgrade, Serbia; nelavilic@gmail.com; 6Clinic of Physical Medicine and Rehabilitation, Clinical Center of Serbia, 11 000 Belgrade, Serbia; 7Department of Anesthesiology, Washington University School of Medicine, St Louis, MO 63110, USA; menelaos.karanikolas@wustl.edu; 8Department of Neurology, Military Medical Academy, 11 000 Belgrade, Serbia

**Keywords:** morphine, transcranial direct current stimulation, acute pain, pain, postoperative, pharmacoeconomics, cost and cost analysis

## Abstract

This prospective randomized study aims to evaluate the feasibility and cost-effectiveness of combining transcranial direct current stimulation (tDCS) with patient controlled intravenous morphine analgesia (PCA-IV) as part of multimodal analgesia after thoracotomy. Patients assigned to the active treatment group (a-tDCS, *n* = 27) received tDCS over the left primary motor cortex for five days, whereas patients assigned to the control group (sham-tDCS, *n* = 28) received sham tDCS stimulations. All patients received postoperative PCA-IV morphine. For cost-effectiveness analysis we used data about total amount of PCA-IV morphine and maximum visual analog pain scale with cough (VASP-C_max_). Direct costs of hospitalization were assumed as equal for both groups. Cost-effectiveness analysis was performed with the incremental cost-effectiveness ratio (ICER), expressed as the incremental cost (RSD or US$) per incremental gain in mm of VASP-C_max_ reduction. Calculated ICER was 510.87 RSD per VASP-C_max_ 1 mm reduction. Conversion on USA market (USA data 1.325 US$ for 1 mg of morphine) revealed ICER of 189.08 US$ or 18960.39 RSD/1 VASP-C_max_ 1 mm reduction. Cost-effectiveness expressed through ICER showed significant reduction of PCA-IV morphine costs in the tDCS group. Further investigation of tDCS benefits with regards to reduction of postoperative pain treatment costs should also include the long-term benefits of reduced morphine use.

## 1. Introduction

The efficacy of transcranial direct current stimulation (tDCS) as an adjuvant nonpharmacological method to conventional regional or systemic analgesia for acute postoperative pain was investigated in seven studies [1,2,3,4,5,6,7]. This was a proof-of-concept clinical trial attempting to explore the impact of tDCS combined with patient controlled intravenous morphine analgesia (PCA-IV) on analgesic use and post-thoracotomy pain. The study is registered in Clinical Trials: https://clinicaltrias.gov (registration number NCT03005548). Two studies explored the cost-effectiveness of tDCS for chronic health problems including treatment of chronic pain [8] and depression [9]. Previously published studies on tDCS efficacy for postoperative pain management tended to focus on opioid consumption and pain intensity, but there was no attempt to explore the cost-effectiveness of tDCS use for acute postoperative pain management.

A cost-effectiveness analysis allows the economic evaluation of a specific medication or technique by measuring costs expressed in currency of the country where analysis is referred to (for example US$) and outcome in natural health units, which indicates an improvement in health such as reduction of pain measured with visual analogue scale in postoperative pain management studies [10]. Cost-effectiveness analysis is an important consideration for expert discussion on new techniques or medications, proposals for health policy, decisions regarding reimbursement in health insurance systems, and also for conducting economic studies regarding the use of novel technology in health care [9,11]. Cost-effectiveness analysis can support the decision for the best treatment matched with the economic feasibility of the health care system when deciding among alternative interventions [11].

tDCS is one of several noninvasive brain stimulation techniques and is getting attention based on its safety level, easy application in hospital, and availability of tDCS devices for home use. Neurophysiological bases for tDCS efficacy is neuromodulation of brain regions known as “the pain matrix” [12], changes in distant connected areas of the brain responsible for motor, cognitive and pain processing [13], increased metabolism, measured by ^18^F-fluorodeoxyglucose, in the cingulate cortex and insula, and decreased metabolism in the left dorsolateral prefrontal cortex [14], as well as the interaction with several neurotransmitters [15].

The analgesic efficacy of multiple tDCS sessions has been explored in chronic pain states, including fibromyalgia [16], and traumatic spinal cord injury [17]. Consequently, as tDCS has been getting attention for acute postoperative pain management, several studies have suggested that the use of tDCS can contribute to reduced postoperative pain intensity and opioid use [1,2,3,5,6,7]. Thoracotomy is a markedly painful incision because it involves cutting through multiple muscle layers, rib resections, intercostal nerve injury and thoracic drain placement at the end of surgery. Consequences of inadequate acute pain management include shallow breathing, inadequate cough strength and expectoration which can contribute to atelectasis and pneumonia [18]. Acute post-thoracotomy pain is the result of several nociceptive and neuropathic mechanisms, and analgesia is usually provided with epidural analgesia or systemic opioids. Inadequately treated post-thoracotomy pain can result in postoperative complications and chronic post-thoracotomy pain [18,19], therefore several different pharmacologic and non-pharmacologic combinations have been used for multimodal analgesia, but they all have side effects. Therefore, the exploration of novel nonpharmacological low-risk techniques is important. We could not find any studies exploring the cost-effectiveness of tDCS applied for acute pain management after thoracotomy.

This study aimed to evaluate the cost-effectiveness of combining tDCS with intravenous morphine patient controlled analgesia (PCA) for the management of acute post-thoracotomy pain. Our second aim was to propose a strategy for future studies integrating tDCS as adjuvant analgesic technique, which will include cost-effectiveness analysis.

## 2. Materials and Methods

This cost-effectiveness study was part of a prospective, randomized, double-blinded sham/controlled proof of concept clinical trial that compared the effectiveness of tDCS as an adjuvant to PCA-IV morphine on analgesia for post-thoracotomy pain. The clinical trial was approved by the Ethics Committee in the Military Medical Academy, Belgrade, Serbia (MF-VMA//December 7th, 2015).

Patients in both groups (active treatment group with tDCS and sham group) received intraoperative intravenous morphine until postoperative visual analogue scale pain in rest (VAS) ≤30 mm was established. The initial tDCS session was followed by postoperative PCA-IV morphine (bolus 1 mg, lockout time 10 min) using the CADD-Legacy PCA Pump (Deltec, Inc., St. Paul, MN, USA). Patients assigned to the active treatment group (a-tDCS, *n* = 27), received tDCS (20 min of 1.2 mA anodal tDCS over the left primary motor cortex for 5 days), whereas patients assigned to the sham control group (sham-tDCS, *n* = 28) received sham tDCS stimulations over the left primary motor cortex for 5 days. Duration (20 min) and location of stimulation (left primary motor cortex) were identical in the tDCS and in the sham stimulation groups.

The primary outcome was the amount of morphine used for analgesia after thoracotomy in a group of patients receiving tDCS in comparison with the amount of morphine used in patients receiving sham stimulation for up to five days. Secondary outcomes were pain scores measured using maximum VAS during cough (VASP-C_max_) in patients receiving tDCS and intravenous morphine PCA, compared to patients receiving sham stimulation and intravenous morphine PCA. Morphine consumption, and pain intensity at rest, with movement, and with cough were recorded at predetermined time intervals as follows: immediately before the intervention, immediately after the intervention, then regularly every one hour for four hours, and then every six hours for five days.

In our study, we analyzed the cost of morphine application in patients with tDCS compared to sham stimulation. The main outcome for the cost-effectiveness analysis was the incremental cost per unit of VAS pain expressed in mm reduction. The main result of cost-effectiveness analysis is the incremental cost-effectiveness ratio (ICER), calculated as the incremental change in costs divided by the incremental change in health outcome [10]. In this cost-effectiveness analysis, the ICER is expressed as the incremental cost (RSD or US$) per incremental gain in natural unit (VASP-C_max_ reduction in mm (1). The prices for morphine were obtained based on data from the official price list in National Health Insurance Fund Republic of Serbia [20] and for the United States of America (USA) market the price list that was available online [21]. For this study, we assumed that direct costs of hospitalization were fixed and equal for both groups. Fixed direct costs included room utilization, equipment acquisition and maintenance, electrodes cost, neurologist and technician coverage for each session, hospitalization costs, anesthesiologist, surgeon, and surgical team fees [8].
(1)$$ICER=\frac{Total\;cost\;of\;morphine\;\left[Sham-Active\right]}{Reduction\;VASP-Cmax\;score\;\left[Sham-Active\right]}$$

The other part of our study included the search of the literature with the topic of tDCS cost utilization for different health problems treatment. Our Medline search included terms (“economics” OR “cost” OR “costs and cost analysis” OR (“costs” AND “cost” AND “analysis”) OR “costs and cost analysis”) AND (“transcranial direct current stimulation” OR (“transcranial” AND “direct” AND “current” AND “stimulation”) OR “transcranial direct current stimulation) revealed 152 abstracts, after reading, two papers were included in analysis [8,9]. The other search including terms (“cost-benefit analysis” OR (“cost-benefit” AND “analysis”) OR “cost-benefit analysis” OR (“cost” AND (“transcranial direct current stimulation” OR (“transcranial” AND “direct” AND “current” AND “stimulation”) OR “transcranial direct current stimulation”) revealed two articles dedicated to tDCS costs. The search was performed on November 17th, 2019, and two independent investigators reviewed abstracts and papers.

## 3. Results

### 3.1. A Cost-Effectiveness Analysis

Data from 27 patients with tDCS stimulation and 28 patients with sham stimulation were included in the analysis. Total morphine dose per group administered during the five days after thoracic surgery was lower in the tDCS group compared to the sham group (2662.2 mg vs. 3518.4 mg). The average morphine use per patient [median (25–75th interquartile range)] was lower in the tDCS group [77.00 (54.00–123.00) mg] compared with the sham group [112.00 (79.97–173.35) mg] and the observed difference is highly significant (*p* = 0.043).

There was no significant difference in VASP-C_max_ between groups on the first postoperative day (in tDCS group 49.00 (40.00–62.00) mm vs. sham group 59.00 (49.00–78.00) mm (*p* = 0.085)). On postoperative day 5, VASP-C_max_ was lower in the tDCS group compared to the sham group [29.00 mm (20.00–39.00) mm vs. 44.50 (30.00–61.75) mm], and the observed difference between the two groups was highly significant (*p* = 0.018). In Serbia, the cost of 1 mg of morphine is 3.58 RSD, and the calculated ICER is 510.87 RSD per 1 mm reduction of VASP-C_max_ (tDCS to sham). Based on this cost estimation, tDCS stimulation is superior to sham as evidenced by reduced cost of analgesic medication used in the tDCS group (Table 1).

The next step was the conversion of the values calculated in RSD to morphine prices in the USA market (Table 2). Our calculation was based on a cost of 1 mg morphine sulfate in the USA of $ 1.325 and resulted in ICER US$ 189.08 or 18,960.39 RSD per 1 mm reduction of VASP-C_max_, which means that morphine cost was reduced by US$ 189.08 with 1 mm reduction of VAS pain. Based on this calculation, the 11mm reduction of VAS maximal pain during cough in the tDCS group resulted in reduction of overall morphine costs by US$ 1134.46 over five days, which means that costs of morphine utilization were lower by US$ 35.85 (3595.05 RSD) per patient in the tDCS group of our study (Table 2).

### 3.2. Literature Search

Our literature search identified two manuscripts published in English, one from the USA and one from France, evaluating the effect of tDCS on utilization costs (Table 3), which present components of tDCS application costs (Table 4).

## 4. Discussion

Cost-effectiveness analysis of tDCS use as an adjuvant non-pharmacologic method in addition to PCA-IV morphine during a five days follow up period showed superiority of tDCS with regards to reduced overall cost for morphine used after thoracotomy, expressed as ICER of 510.87 RSD per 1 mm reduction in VASP-C_max_ in the tDCS group compared to sham in Serbia. Conversion of the data to USA market prices revealed ICER US$ 189.08 or 18,960.39 RSD per 1 mm VASP-C_max_ reduction. To our knowledge, our study is the first cost-effectiveness analysis performed on the effects of tDCS on the cost of intravenous morphine analgesia for acute postoperative pain management.

The reason for including VASP-C_max_ at day five in the analysis was based on knowledge about cumulative tDCS effect. The repeated tDCS sessions result in cumulative tDCS effect as consequence of cumulative and sustained changes in cerebral function [22,23]. Studies exploring tDCS effects in acute postoperative pain management have shown a reduction of postoperative opioid use that ranges from 22% for hydromorphone [1] to 73.25% for other analgesics [7]. Pain intensity is reduced depending on the type of surgical procedure and patient population and it is measured and presented differently in different studies. Our literature search revealed seven studies utilizing tDCS for acute postoperative pain, but only three of them presented results on opioid use and pain measurement [2,4,6]. The model used in this study highlights the increased cost of pain management when pain intensity increases. However, it was difficult to apply the ICER methodology calculation using data from previous studies, because these studies differ in the type of data presentation (average opioid use), duration of follow up, type of opioid, and country of origin. Therefore, we suggest that future studies should state the total amount of opioids in active and sham tDCS groups and VAS or another type of scale measured pain for particular measurement time points. In this way, data from multiple studies can provide accurate information about the cost-effectiveness of tDCS as an adjuvant nonpharmacologic method to postoperative analgesia.

Postoperative pain is a complex phenomenon, and despite the plethora of pharmacological and non-pharmacological approaches, adequate pain management is still challenging in the hospital and even more so after hospital discharge [24,25]. Inadequate pain relief complicates postoperative recovery with increased morbidity, functional impairment, quality-of-life impairment, delayed recovery time, prolonged duration of opioid use, increased length of stay, readmission rates, and higher health-care costs [25,26]. In the US, the annual cost to society of common chronic pain conditions, including postoperative pain, was conservatively estimated to be in the range of US$560-635 billion [27]. Therefore, pain after surgery represents a financial burden for healthcare systems worldwide, especially for developing countries like Serbia [28,29,30,31]. The prediction is that the number of surgical procedures will keep increasing in the future, and the higher number of surgical procedures could aggravate existing problems related to the long term postoperative use of opioid and nonopioid analgesics and adjuvants [32,33,34,35]. Therefore, novel analgesia techniques, such as tDCS, which have already shown opioid-sparing effects in the postoperative period and efficacy of drug addiction, treatment can be very beneficial [36].

Among studies that explored cost-effectiveness on tDCS use, Zaghi et al. calculated the price of tDCS application in chronic pain treatment as US$ 167.72 per session [8]. In their thorough analysis of tDCS cost for depression treatment, Sauvaget et al. presented data including device cost (ranges from 8000 to 14,000 euros), and additional costs such as maintenance cost, staff training, staff (neurologist and technician) performance at the site [9]. However, the authors point out that the price of the tDCS session decreases with the number of patients and applications [9]. Although tDCS is used for depression treatment in our hospital for the last seven years, the national health insurance fund in Serbia does not recognize it for reimbursement to the hospital. Preclinical identification of responders to tDCS effects could be helpful for clinical and for economic reasons [37] and might be useful for the management of postoperative pain, particularly in patients with preexisting pain, while avoiding the use of tDCS in non-responders, thereby avoid wasting resources including costs of tDCS application.

Our cost-effectiveness analysis has several limitations. The findings might have been more convincing if we had included a larger number of patients from several hospitals. However, this was a feasibility study, and we considered the cost of morphine but did not include the cost of tDCS, because we could not find any data on tDCS cost in Serbian health reimbursement’s lists. Furthermore, we did not include in the analysis “fixed direct costs” as defined by Zaghi et al. because our study was a single-center study and all patients were treated in the same ward [8]. Therefore, our analysis is based on the assumption of the equal costs of other elements than morphine cost. None of the studies investigating tDCS efficacy on postoperative pain opioid use and pain intensity provide sufficient data to perform cost-effectiveness analysis. However, based on available literature, we prepared a proposal for the data necessary to enable post hoc cost-effectiveness analysis. This proposal needs further review and corrections in preparation for its application.

Published studies on tDCS application for acute postoperative pain have focused on short term tDCS effect, and there could be an argument that based on the cost of tDCS as calculated by Zaghi et al. [8], use of tDCS is not cost effective, because the observed reduction of morphine use does not justify the added cost. However, it is important to note that the benefit of reduced morphine use, in addition to lower pharmacy costs, also includes a reduced risk of opioid dependence, lower incidence of chronic pain, improved quality of recovery, and improved quality of life. These additional benefits are not part of the cost to benefit analysis in our study, but we believe that these more difficult to measure outcomes need to be investigated in future research on this topic, which could include long term postoperative follow up.

A reasonable approach to tDCS use for postoperative pain management could include the application of portable tDCS devices for home use [38], which can be supported by staff adequately trained in remote patient follow up and collection of data for assessment of treatment efficacy [39]. Published studies support the benefit and cost-effectiveness of tDCS as an adjuvant non-pharmacologic technique for acute postoperative pain, therefore future revised guidelines on postoperative pain management could perhaps also include neuromodulation techniques as part of multimodal analgesia therapies. We suggest that authors provide necessary data for cost-effectiveness analysis calculations in the appendix or supplement of future studies, to facilitate future post hoc cost-effectiveness analysis (Table 4).

## 5. Conclusions

Our study showed significant, based on ICER calculation, reduction of morphine costs in patients who received three to five repeated tDCS sessions, compared to patients who received sham treatments for acute pain after thoracotomy. The cost-effectiveness analysis is based on the assumption of the equal costs of other than morphine.

This result suggests that future studies should perhaps provide data for post hoc cost-effectiveness analysis because such data could help clinicians and health care decision-makers to better allocate available resources for the treatment of postoperative pain.

## Figures and Tables

**Table 1 ijerph-17-00816-t001:** Utility and expenditure of morphine cost in patients where active or sham transcranial direct current stimulation was used for post-thoracotomy pain treatment.

1 mg morphine = 3.58 RSD 1 US$ = 100.2784 RSD	Type of Stimulation	
	Active Group (*n* = 27)	Sham Group (*n* = 28)
Cumulative morphine	2662.2 mg	3518.4 mg
VASP-C_max_ on postoperative day 1 *(A)	40.00 mm	49.50 mm
VASP-C_max_ on postoperative day 5 (B)	29.00 mm	44.50 mm
Reduction VASP-C_max_ score (A minus B)	11 mm	5 mm
Incremental cost-effectiveness ratio (ICER)	510.87 RSD/1 VASP-C_max_ mm reduction	
Morphine cost per patient	352.99 RSD	449.85 RSD

VASP-C_max_-maximum visual analogue pain score with cough, RSD-Republic of Serbia Dinar; US$-United States Dollar; tDCS-transcranial direct current stimulation; A- VASP-C_max_ on postoperative day 1; B- VASP-C_max_ on postoperative day 5.

**Table 2 ijerph-17-00816-t002:** Utility and expenditure of morphine cost data conversion from Serbian market to United States of America (USA) market, in the study population of patients where active or sham transcranial direct current stimulation was used for post-thoracotomy pain treatment.

1 mg morphine = 1.325 US$ 1 US$ = 100.2784 RSD	Type of Stimulation	
	Active Group (*n* = 27)	Sham Group (*n* = 28)
Cumulative morphine	2662.2 mg	3518.4 mg
VASP-C_max_ on postoperative day 1 (A)	40.00 mm	49.50 mm
VASP-C_max_ on postoperative day 5 (B)	29.00 mm	44.50 mm
Reduction VASP-C_max_ (A minus B)	11 mm	5 mm
Incremental cost-effectiveness ratio (ICER)	18,960.39 RSD/1 VASP-C_max_ mm reduction	
Morphine cost per patient	13,100.87 RSD	16,695.92 RSD

VASP-C_max_-maximum visual analogue pain score with cough, RSD-Republic of Serbia Dinar; US$-United States Dollar; tDCS-transcranial direct current stimulation; A- VASP-C_max_ on postoperative day 1; B- VASP-C_max_ on postoperative day 5.

**Table 3 ijerph-17-00816-t003:** Published data on the use and cost of transcranial direct current stimulation (tDCS) for treatment of chronic medical conditions. Data are available only on tDCS for chronic pain (Zaghi et al., 2009) [8] and depression (Sauvaget et al., 2019) [9].

Medical Condition	Price of tDCS per Session US$ (€)	Number of Patients (N)	Year of Publication Country of Data Origin	Costs
Chronic pain [8]	167.72 (US$)	22	2009 USA	Room utilization Equipment maintenance, supplies Technician Neurologist consultation Administrative fees Hospitalization Surgeon’s fee Anesthesiologist’s fee Surgical fee Electrodes
Depression [9]	103.71 (EU€)	N/A	2019 France	Equipment Staff Structural costs

US$-United States Dollar, EU€-European Union euro, USA-United States of America

**Table 4 ijerph-17-00816-t004:** The ”impact inventory” table for transcranial direct current (tDCS) stimulation cost-effectiveness analysis for acute postoperative pain management. The concept of “impact inventory” was suggested by Neumann et al. [11], and the necessary information was retrieved from Zaghi et al. [8] and (Sauvaget et al., 2019) [9].

Specific Parameter	Component
Room utilization	Country and hospital dependent costs
Administrative fees	Country and hospital dependent costs
Hospitalization costs	Country and hospital dependent costs Room utilization, surgical team fee, procedure costs
Equipment costs	Device price Equipment maintenance, supplies Electrode costs
Staff costs	Country and hospital dependent Technician Neurologist consultation Surgeon fee Anesthesiologist fee Surgical fee
Basic analgesia technique cost	PCA/PCEA pumps Infusion system costs Medications price Epidural/peripheral nerve block set costs
Clinical benefit	Opioid reduction Pain intensity reduction

PCA-patient controlled analgesia, PCEA-patient controlled epidural analgesia.

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
