# Peer review of "Patient-Controlled Intravenous Morphine Analgesia Combined with Transcranial Direct Current Stimulation for Post-Thoracotomy Pain: A Cost-Effectiveness Study and A Feasibility for Its Future Implementation"

_ijerph, 2020, doi:10.3390/ijerph17030816_

Round 1

Reviewer 1 Report

I have the following concerns:

You do not include costs other than that of morphine into the calculation. You pretend that tDCS is free of charge. Extrapolation to US currency is not possible, as you do not have the same costs of room rental, salaries, etc. As you wrote, the incremental cost should be computed per unit of natural measure. So it should be in RSD per one mm of pain reduction [VAS 0-100 mm], not per one percent or VAS.

Other comments are included in the file attached.

I do not refer to the discussion section, as it is a consequence of your assumptions and methodology that I have meaningful concerns about.

Author Response

We greatly appreciate your comments, and we also want to sincerely thank you for time and effort.

Comments for authors:

You do not include costs other than that of morphine into the calculation. You pretend that tDCS is free of charge.

In response:

In the discussion section we described the problem of tDCS cost in Serbia as follows:

Our cost-effectiveness analysis has several limitations. The findings might have been more convincing, if we had included a larger number of patients, from several hospitals. However, this was a feasibility study, and we considered the cost of morphine, but did not include the cost of tDCS, because we could not find any data on tDCS cost in Serbian health reimbursement’s lists.

Extrapolation to US currency is not possible, as you do not have the same costs of room rental, salaries, etc.

Based on your comment we exchanged the word “extrapolation” with the word “conversion”

3.As you wrote, the incremental cost should be computed per unit of natural measure. So it should be in RSD per one mm of pain reduction [VAS 0-100 mm], not per one percent or VAS.

In response to the above comment, we reanalyzed the data and changed the following tables and sentences:

Data from 27 patients with tDCS stimulation and 28 patients with sham stimulation were included in the analysis. Total morphine dose per group administered in each group during the five days after thoracic surgery was lower in the tDCS group compared to the sham group (2662.2 mg vs. 3518.4 mg). The average morphine use per patient was lower in the tDCS group [77.00 (54.00-123.00) mg] compared with the sham group [112.00 (79.97-173.35) mg] and the observed difference is highly significant (p=0.043).

       On postoperative day 5, VASP-Cmax was lower in the tDCS group compared to the sham group [29.00 mm (20.00-39.00) mm vs. 44.50 (30.00-61.75) mm, and the observed difference between the two groups was highly significant (p=0.018). In Serbia, the cost of 1 mg of morphine is 3.58 RSD, and the calculated ICER) is 510.87 RSD per 1 mm reduction of VASP-Cmax [tDCS to sham]. Based on this cost estimation, tDCS stimulation is superior to sham as evidenced by reduced cost of analgesic medication used in the tDCS group (Table 1).

Table 1. Utility and expenditure of morphine cost in patients where active or sham transcranial direct current stimulation was used for post-thoracotomy pain treatment.

1 mg morphine = 3.58 RSD

1 US$ = 100.2784 RSD

Type of stimulation

Active group

(n=27)

Sham group

(n=28)

Cumulative morphine

2662.2 mg

3518.4 mg

VASP-Cmax on postoperative day 1 a

40.00 mm

49.50 mm

VASP-Cmax on postoperative day 5b

29.00 mm

44.50 mm

Reduction VASP-Cmax score (a - b)

11 mm

5 mm

Incremental cost-effectiveness ratio (ICER)

510.87 RSD/1 VASP-Cmax mm reduction

VASP-Cmax -maximum visual analogue pain score with cough, RSD-Republic of Serbia Dinar; US-United States Dollar; tDCS-transcranial direct current stimulation

The next step was a conversion of the values calculated in RSD to morphine prices in the USA market (Table 2). Our calculation was based on cost of 1 mg morphine sulfate in the USA of $ 1.325 and resulted in ICER US$ 189.08 or 18960.39 RSD per 1 VASP-Cmax mm reduction, which means that morphine cost was reduced by US$ 189.08 with one mm VAS pain reduction. Based on this calculation, the 11mm reduction of VAS maximal pain during cough in the tDCS group resulted in reduction of overall morphine costs by US$ 1134.46 over five days, which means that costs of morphine utilization were lower by US$ 35.57 per patient in the tDCS group of our study.

Table 2. Utility and expenditure of morphine cost data conversion from the Serbian market on the United States of America (USA) market in the study population of patients where active or sham transcranial direct current stimulation was used for post-thoracotomy pain treatment.

1 mg morphine = 1.325 US$

1 US$ = 100.2784 RSD

Type of stimulation

Active group

(n=27)

Sham group

(n=28)

Cumulative morphine

2662.2 mg

3518.4 mg

VASP-Cmax on postoperative day 1a

40.00 mm

49.50 mm

VASP-Cmax on postoperative day 5b

29.00 mm

44.50 mm

Reduction VASP-Cmax (a - b)

11 mm

5 mm

Incremental cost-effectiveness ratio (ICER)

18960.39 RSD/1 VASP-Cmax mm reduction

VASP-Cmax -maximum visual analogue pain score with cough, RSD-Republic of Serbia Dinar; US-United States Dollar; tDCS-transcranial direct current stimulation

P2-Line 47 None of those studies concerned post-thoracotomy pain

In response we added the following sentence in the revised manuscript we are now submitting:

This was a proof-of-concept clinical trial attempting to explore the impact of tDCS combined with patient controlled intravenous morphine analgesia on analgesic use and post-thoracotomy pain. The study is registered in Clinical Trials: https://clinicaltrias.gov (registration number NCT03005548).

5.P2-Line 50 How about reference number 8?

In response:

P2-line (50-52) In section Introduction:

Two studies explored the cost-effectiveness of tDCS for chronic health problems, including treatment of chronic pain [8] and depression [9].

The reference number 8 (Zaghi, S.; Heine, N.; Fregni, F. (2009). Brain stimulation for the

treatment of pain: A review of costs, clinical effects, and mechanisms of treatment for three

different central neuromodulatory approaches. J Pain Manag. 2009, 2, 339-350.) is dedicated to

chronic pain not the management of acute postoperative pain.

In response, we added the following words in the sentence:

Previously published studies on tDCS efficacy for postoperative pain management tended to focus on opioid consumption and pain intensity, but there was no attempt to explore the cost-effectiveness of tDCS use for acute postoperative pain management.

6.P2-Line 72 nociceptive and neuropathic mechanisms

In response, we added the following words in the sentence:

Acute post-thoracotomy pain is the result of several nociceptive and neuropathic mechanisms, can be very intense, and is more effectively treated with multimodal analgesia regimens in combination with conventional analgesia provided with epidural analgesia or systemic opioids.

7.P3-Line 113-115 I think you can not do it

The term fixed direct costs include room utilization, equipment price and maintaining, electrodes cost, neurologist and technician coverage for each session, hospitalization costs, anesthesiologist, surgeon and surgical team fees [8].

In response:

In our clinical trial both patient groups were treated in the same system and hospital, which is the same principle used in the study performed by Zaghi et al. [8].

In response, we rewrote the sentences (also based on a suggestion by Reviewer#3):

For this study, we assumed that the direct costs of hospitalization were fixed and equal for both groups. The fixed direct costs included room utilization, equipment price and maintaining, electrodes cost, neurologist and technician coverage for each session, hospitalization costs, anesthesiologist, surgeon and surgical team fees [8].

In response, we added the following sentence in the section Discussion

Furthermore, we did not include in the analysis “fixed direct costs” as defined by Zaghi et al. because our study was a single-center study and all patients were treated in the same ward [8].

8.Line 135 Is that difference statistically different? (2662.2 mg vs. 3518.4 mg)

In response

The total morphine dose for the tDCS group (2662.2 mg) vs. the total morphine dose in the sham group (3518.4 mg) is usually not compared for statistical significance. However, we can compare the average amount of morphine per patient and this comparison showed significantly lower morphine use in the tDCS group [77.00 (54.00-123.00) mg] compared with the sham group [112.00 (79.97-173.35) mg, p=0.043].

The following sentence was inserted:

The average morphine use per patient was lower in the tDCS group [77.00 (54.00-123.00) mg] compared with the sham group [112.00 (79.97-173.35) mg] and the observed difference is highly significant (p=0.043).

9.Line 137 Is it statistically different? p? (29.00 mm vs. 44.50 mm)

In response, we added the following data:

On postoperative day 5, VASP-Cmax was lower in the tDCS group compared to the sham group [29.00 mm (20.00-39.00) mm vs. 44.50 (30.00-61.75) mm, and the observed difference between the two groups was highly significant (p=0.018).

Reviewer 2 Report

Congrats for your excelent work. I´ll tell you some considerations I think yout work must be improve.

Abstract: Structure it usual (aims, methods, results, conclusión). Key words: must be found in MesH (pubMed)

Introduction: I think you must explain factors in postoperative thoracic pain and how can it be treat.

Methods. 

Results: It must be resume in a table

Author Response

We greatly appreciate your comments, and we also want to sincerely thank you for time and effort.

1)Abstract: Structure it usual (aims, methods, results, conclusión).

In response:

The Abstract is written as is based on MDPI instructions for authors, which require that the abstract should be structured by content, without headings: “Abstract: The abstract should be a total of about 200 words maximum. The abstract should be a single paragraph and should follow the style of structured abstracts, but without headings.”

2)Key words: must be found in MesH (pubMed)

In response:

We used the MeSH terms for key words except for “post-thoracotomy pain” as we thought this is an important issue. Based on reviewer’s suggestion we added the following MeSH terms pain, postoperative

3)Introduction: I think you must explain factors in postoperative thoracic pain and how can it be treat.

In response we added the sentence in section Introduction:

Thoracotomy is a markedly painful incision because it involves cutting through multiple muscle layers, rib resections, intercostal nerve injury and thoracic drain placement at the end of surgery. Consequences of inadequate acute pain management include shallow breathing, inadequate cough strength and expectoration which can contribute to atelectasis and pneumonia  [18]. Acute post-thoracotomy pain is the result of several nociceptive and neuropathic mechanisms, and analgesia is usually provided with epidural analgesia or systemic opioids. Because inadequately treated post-thoracotomy pain can result in postoperative complications and chronic post-thoracotomy pain [18,19], several different pharmacologic and non-pharmacologic combinations have been used for multimodal analgesia, but they all have side effects. Therefore, exploration of novel nonpharmacological low-risk techniques is important. We could not find any studies exploring the cost-effectiveness of tDCS applied for acute pain management after thoracotomy.

Reviewer 3 Report

line 45 no comma after (tDCS)

sentence starting line 55-58 seems too long or not quite correctly punctuated (possibly just missing a comma somewhere in line 58)

line 59 "can support the decision on the best treatment" should probably be "can support the decision for the best treatment"

line 63 perhaps "the 'pain matrix'"?

line 95 please specify whether duration & location of sham treatments was identical to those of the active treatment (I assume they were)

line 114 perhaps "were fixed and equal" rather than "are fixed and same". and perhaps "The fixed direct costs included" rather than "The term fixed direct costs include"

I would be really interested to know if there was any difference in length of stay between groups, though this study may not have been powered to detect such a difference, or data on length of stay may not have been collected. Also, oral opioid use after discontinuation of the PCA and discharge opioid requirements would be of definite interest!

lines 120-130 Not sure if literature review for cost-effectiveness of tDCS is really a part of the study... but definitely appreciate the thoroughness of search terms.

Table 1: consider "postoperative day 1" and "postoperative day 5" for ease of reading/interpretation. also, please consider further clarifying that total morphine dose is the total for the group, not per patient.

I suspect the a-tDCS had a small cost savings in PCA syringes as well, though this would be somewhat more difficult to measure, and may be slightly safer by reducing time of continuous IV access, thereby marginally reducing risk for bloodstream infections.

line 147 & 162: "VAS-C" should be "VASP-C"

Are the contents of Tables 1 & 2 reversed? (According to their titles, Table 1 = ICER in RSD; Table 2 = ICER in US$)

line 164: should be section 3.2 if including literature search as a study component.

line 180: consider "showed superiority in terms of" or "showed superiority based on"

Good discussion of limitations, especially lack of tDCS cost. It might be helpful to contact the hospital at which the study was performed, or the neurology department, to try to estimate costs. 

Overall, this is a really nice feasibility study, and I very much hope you will proceed with larger studies to further demonstrate whether tDCS is truly cost-effective. I think it shows great promise based on your data--even if it were equal in cost, the improved pain control is an important outcome!

Author Response

We greatly appreciate your comments, and we also want to sincerely thank you for time and effort.

1)line 45 no comma after (tDCS)

In response to the reviewer’s suggestion comma is deleted (Line 45)

The efficacy of transcranial direct current stimulation (tDCS) as an adjuvant nonpharmacological method to conventional regional or systemic analgesia for acute postoperative pain was investigated in seven studies [1–7].

2)sentence starting line 55-58 seems too long or not quite correctly punctuated (possibly just missing a comma somewhere in line 58)

In response to the reviewer’s suggestion, we rewrote the sentence:

Cost-effectiveness analysis is an important consideration for expert discussion on new techniques or medications, proposals for health policy, decisions regarding reimbursement in health insurance systems and also for conducting economic studies regarding the use of novel technology in health care [9,11].

3)line 59 "can support the decision on the best treatment" should probably be "can support the decision for the best treatment"

In response to the reviewer’s suggestion, we corrected the following sentence:

Cost-effectiveness analysis can support the decision for the best treatment matched with the economic feasibility of the health care system when deciding among alternative interventions [11].

4)line 63 perhaps "the 'pain matrix'"?

In response to reviewer’s suggestion we corrected the following sentence:

Neurophysiological bases for tDCS efficacy is neuromodulation of brain regions known as “the pain matrix”

5)line 95 please specify whether duration & location of sham treatments was identical to those of the active treatment (I assume they were)

In response to reviewer’s comment the following sentence was added in the section Materials and Methods:

Duration (20 minutes) and location of stimulation (left primary motor cortex) were identical in the tDCS and in the sham stimulation groups.

6)line 114 perhaps "were fixed and equal" rather than "are fixed and same". and perhaps "The fixed direct costs included" rather than "The term fixed direct costs include"

In response to reviewer’s suggestion we corrected the following sentence:

For the purpose of this study, we assumed that direct costs of hospitalization were fixed and equal for both groups.

Fixed direct costs included room utilization, equipment acquisition and maintenance, electrodes cost, neurologist and technician coverage for each session, hospitalization costs, anesthesiologist, surgeon and surgical team fees [8].

7)I would be really interested to know if there was any difference in length of stay between groups, though this study may not have been powered to detect such a difference, or data on length of stay may not have been collected. Also, oral opioid use after discontinuation of the PCA and discharge opioid requirements would be of definite interest! 

In response

Our study was focused on the impact of tDCS on morphine use during five postoperative days. Therefore, we did not include follow up data on postoperative opioid use and hospital stay.

8)lines 120-130 Not sure if literature review for cost-effectiveness of tDCS is really a part of the study... but definitely appreciate the thoroughness of search terms.

9)Table 1: consider "postoperative day 1" and "postoperative day 5" for ease of reading/interpretation.

In response to the reviewer’s suggestion the following changes were inserted in the tables:

Table 1. Utility and expenditure of morphine cost in patients where active or sham transcranial direct current stimulation was used for post-thoracotomy pain treatment.

1 mg morphine = 3.58 RSD

1 US$ = 100.2784 RSD

Type of stimulation

Active group

(n=27)

Sham group

(n=28)

Cumulative morphine

2662.2 mg

3518.4 mg

VASP-Cmax on postoperative day 1 a

40.00 mm

49.50 mm

VASP-Cmax on postoperative day 5b

29.00 mm

44.50 mm

Reduction VASP-Cmax score (a - b)

11 mm

5 mm

Incremental cost-effectiveness ratio (ICER)

510.87 RSD/1 VASP-Cmax mm reduction

VASP-Cmax -maximum visual analogue pain score with cough, RSD-Republic of Serbia Dinar; US-United States Dollar; tDCS-transcranial Direct Current Stimulation

Table 2. Utility and expenditure of morphine cost data conversion from the Serbian market on the United States of America (USA) market, in the study population of patients where active or sham transcranial direct current stimulation was used for post-thoracotomy pain treatment.

1 mg morphine = 1.325 US$

1 US$ = 100.2784 RSD

Type of stimulation

Active group

(n=27)

Sham group

(n=28)

Cumulative morphine

2662.2 mg

3518.4 mg

VASP-Cmax on postoperative day 1a

40.00 mm

49.50 mm

VASP-Cmax on postoperative day 5b

29.00 mm

44.50 mm

Reduction VASP-Cmax (a - b)

11 mm

5 mm

Incremental cost-effectiveness ratio (ICER)

18960.39 RSD/1 VASP-Cmax mm reduction

VASP-Cmax -maximum visual analogue pain score with cough, RSD-Republic of Serbia Dinar; US-United States Dollar; tDCS-Transcranial Direct Current Stimulation

also, please consider further clarifying that total morphine dose is the total for the group, not per patient.

In response to the reviewer’s suggestion, the following words are added in the sentence:

Total morphine dose per group administered during the five days after thoracic surgery was lower in the tDCS group compared to the sham group (2662.2 mg vs. 3518.4 mg).  

10)I suspect the a-tDCS had a small cost savings in PCA syringes as well, though this would be somewhat more difficult to measure, and may be slightly safer by reducing time of continuous IV access, thereby marginally reducing risk for bloodstream infections.

In response: We thank the Reviewer for this consideration about cost savings on syringes; however, the pumps CADD­Legacy PCA Pump (Deltec, Inc., St.Paul, MN, USA) used for this study only accept containers (volume 100 ml), and the syringes are rarely used.

11)line 147 & 162: "VAS-C" should be "VASP-C"

In response the following changes were done:

The following word was writing mistake and it is deleted

12)Are the contents of Tables 1 & 2 reversed? (According to their titles, Table 1 = ICER in RSD; Table 2 = ICER in US$)

In response to the reviewer:

Table 1 presents analysis based on morphine prices in Serbia. Table 2 presents analysis of the prices of medication in US. We used this to make data understandable for a wider reader audience.

13)line 164: should be section 3.2 if including literature search as a study component.

In response to the reviewer’s comment, the following correction is inserted:

3.2. Literature search

14) line 180: consider "showed superiority in terms of" or "showed superiority based on"

In response to the reviewer’s comment following change was done:

Cost-effectiveness analysis of tDCS use as an adjuvant non-pharmacologic method in addition to PCA-IV morphine during a five days follow up period showed superiority of tDCS with regards to reduced overall cost for morphine used after thoracotomy…

15)A good discussion of limitations, especially lack of tDCS cost. It might be helpful to contact the hospital at which the study was performed, or the neurology department, to try to estimate costs. 

In response to reviewer’s comments:

The Department of Neurology has an overall budget which is part of a larger Hospital budget, but does not track details about every cost item in their budget. Therefore, it was not possible to obtain detailed cost estimates related to the use of tDCS in this study.

Round 2

Reviewer 1 Report

My major concerns are:

You should clearly state in the abstract and in the discussion that your cost-effectiveness analysis is based on the assumption of the equal costs of other than morphine means. Otherwise, the results and conclusions remain misleading. There is no information if there is any statistical difference between values for VASP-Cmax in the active and sham groups on postoperative day 1. The statistical calculations should be performed on a patient level, not on means.

Author Response

Thank you for taking the time to review our manuscript. After carefully going over your suggestions and comments, we believe we can address most of these concerns in a satisfactory manner. Therefore, we are now submitting a revised manuscript for your consideration.

Comments for authors:

1)You should clearly state in the abstract and in the discussion that your cost-effectiveness analysis is based on the assumption of the equal costs of other than morphine means. Otherwise, the results and conclusions remain misleading.

In response:

In the abstract section we inserted the following sentence:

Direct costs of hospitalization were assumed as equal for both groups.

In the discussion we inserted the following:

Therefore, our analysis is based on the assumption of the equal costs of other elements than morphine cost.

The following sentence is added in the conclusion

The cost-effectiveness analysis is based on the assumption of the equal costs of other than morphine.

2)There is no information if there is any statistical difference between values for VASP-Cmax in the active and sham groups on postoperative day 1.

In response, we added the following sentence in section Results:

There was no significant difference in VASP-Cmax between groups on the first postoperative day (in tDCS group 49.00 (40.00-62.00) mm vs. sham group 59.00 (49.00-78.00) mm (p=0.085)).

In response, we added the following sentence in the section Discussion:

The reason for including only VASP-Cmax at day five in the analysis was based on knowledge about cumulative tDCS effect. The repeated tDCS sessions result in cumulative tDCS effect as consequence of cumulative and sustained changes in cerebral function (Mori et al., 2010, Alonzo et al.,2012).

Alonzo, A., Brassil, J., Taylor, J.L., Martin, D., and Loo, C.K. (2012). Daily transcranial direct current stimulation (tDCS) leads to greater increases in cortical excitability than second daily transcranial direct current stimulation. Brain Stimul  5:208-213. doi: 10.1016/j.brs.2011.04.006.

Mori, F., Codeca, C., Kusayanagi, H., Monteleone, F., Buttari, F., Fiore, S., et al. (2010). Effects of anodal transcranial direct current stimulation on chronic neuropathic pain in patients with multiple sclerosis. J Pain  11:436-442. doi: 10.1016/j.jpain.2009.08.011.

3)The statistical calculations should be performed on a patient level, not on means. 

In response to reviewer’s request the changes were made in the tables and the body text.

Based on this calculation, the 11mm reduction of VAS maximal pain during cough in the tDCS group resulted in reduction of overall morphine costs by US$ 1134.46 over five days, which means that costs of morphine utilization were lower by US$ 35.85 (3595.05 RSD) per patient in the tDCS group of our study (Table 2).

Table 1. Utility and expenditure of morphine cost in patients where active or sham transcranial direct current stimulation was used for post-thoracotomy pain treatment.

1 mg morphine = 3.58 RSD

1 US$ = 100.2784 RSD

Type of stimulation

Active group

(n=27)

Sham group

(n=28)

Cumulative morphine

2662.2 mg

3518.4 mg

VASP-Cmax on postoperative day 1a

40.00 mm

49.50 mm

VASP-Cmax on postoperative day 5b

29.00 mm

44.50 mm

Reduction VASP-Cmax score (a - b)

11 mm

5 mm

Incremental cost-effectiveness ratio (ICER)

510.87 RSD/1 VASP-Cmax mm reduction

Morphine cost per patient

352.99 RSD

449.85 RSD

VASP-Cmax -maximum visual analogue pain score with cough, RSD-Republic of Serbia Dinar; US-United States Dollar; tDCS-transcranial direct current stimulation

Table 2. Utility and expenditure of morphine cost data conversion from Serbian market on United States of America (USA) market, in the study population of patients where active or sham transcranial direct current stimulation was used for post-thoracotomy pain treatment.

1 mg morphine = 1.325 US$

1 US$ = 100.2784 RSD

Type of stimulation

Active group

(n=27)

Sham group

(n=28)

Cumulative morphine

2662.2 mg

3518.4 mg

VASP-Cmax on postoperative day 1a

40.00 mm

49.50 mm

VASP-Cmax on postoperative day 5b

29.00 mm

44.50 mm

Reduction VASP-Cmax (a - b)

11 mm

5 mm

Incremental cost-effectiveness ratio (ICER)

18960.39 RSD/1 VASP-Cmax mm reduction

Morphine cost per patient

13100.87 RSD

16695.92 RSD

VASP-Cmax -maximum visual analogue pain score with cough, RSD-Republic of Serbia Dinar; US-United States Dollar; tDCS-transcranial direct current stimulation

Reviewer 2 Report

I think the study can be improved if you insert and structure abstract (aims, methods, results, conclusions).

Congrats!

Author Response

Thank you for taking the time to review our manuscript. After carefully going over your suggestion, we believe we can addressed your concern in a satisfactory manner. Therefore, we are now submitting a revised manuscript for your consideration.

1)I think the study can be improved if you insert and structure abstract (aims, methods, results, conclusions).

In response to the reviewer’s comment:

The Abstract is written as is based on MDPI instructions for authors, which require that the abstract should be structured by content, without headings: “Abstract: The abstract should be a total of about 200 words maximum. The abstract should be a single paragraph and should follow the style of structured abstracts, but without headings.” Therefore, the abstract inserted in the manuscript does not include headings.

Abstract:Aims: This prospective randomized study aims to evaluate the feasibility and cost-effectiveness of combining transcranial direct current stimulation (tDCS) with PCA-IV morphine as part of multimodal analgesia after thoracotomy. Methods: Patients assigned to the active treatment group (a-tDCS, n=27) received tDCS over the left primary motor cortex for five days, whereas patients assigned to the control group (sham-tDCS, n=28) received sham tDCS stimulations. All patients received postoperative PCA-IV morphine. For cost-effectiveness analysis we used data about total amount of PCA-IV morphine and maximum visual analog pain scale with cough (VASP-Cmax). Direct costs of hospitalization were assumed as equal for both groups. Cost-effectiveness analysis was performed with the incremental cost-effectiveness ratio (ICER), expressed as the incremental cost (RSD or US$) per incremental gain in mm of VASP-Cmax reduction. Results: Calculated ICER was 510.87 RSD per VASP-Cmax 1 mm reduction. Conversion on USA market (USA data 1.325 US$ for 1 mg of morphine) revealed ICER of 189.08 US$ or 18960.39 RSD/1 VASP-Cmax 1 mm reduction. Conclusions: Cost-effectiveness expressed through ICER showed significant reduction of PCA-IV morphine costs in the tDCS group. Further investigation of tDCS benefits with regards to reduction of postoperative pain treatment costs should also include the long-term benefits of reduced morphine use.
